# Dynamic phase-locking states and personality in sub-acute mild traumatic brain injury: An exploratory study

Harm J. van der Horn[1,2]*, Myrthe E. de Koning[3], Koen Visser[1], Marius G. J. Kok[4], Jacoba M. Spikman[5], Myrthe E. Scheenen[5], Remco J. Renken[6], Vince D. Calhoun[7], Victor M. Vergara[7], Joana Cabral[8], Andrew R. Mayer[2,9,10,11], Joukje van der Naalt[1]

1 Department of Neurology, University Medical Center Groningen, University of Groningen, Groningen, The Netherlands, 2 The Mind Research Network/Lovelace Biomedical Research Institute, Pete & Nancy Domenici Hall, Albuquerque, NM, United States of America, 3 Department of Neurology, Medical Spectrum Twente, Enschede, The Netherlands, 4 Department of Radiology, University Medical Center Groningen, University of Groningen, Groningen, The Netherlands, 5 Department of Neuropsychology, University of Groningen, University Medical Center Groningen, Groningen, The Netherlands, 6 Department of Neuroscience, University Medical Center Groningen, University of Groningen, Groningen, The Netherlands, 7 Tri-institutional Center for Translational Research (TReNDS), Georgia State, Georgia Tech, Emory, Atlanta, GA, United States of America, 8 Life and Health Sciences Research Institute (ICVS), School of Medicine, University of Minho, Braga, Portugal, 9 Department of Neurology, University of New Mexico School of Medicine, Albuquerque, NM, United States of America, 10 Department of Psychiatry, University of New Mexico School of Medicine, Albuquerque, NM, United States of America, 11 Department of Psychology, University of New Mexico School of Medicine, Albuquerque, NM, United States of America

* h.j.van.der.horn@umcg.nl

## Abstract

Research has shown that maladaptive personality characteristics, such as Neuroticism, are associated with poor outcome after mild traumatic brain injury (mTBI). The current exploratory study investigated the neural underpinnings of this process using dynamic functional network connectivity (dFNC) analyses of resting-state (rs) fMRI, and diffusion MRI (dMRI). Twenty-seven mTBI patients and 21 healthy controls (HC) were included. After measuring the Big Five personality dimensions, principal component analysis (PCA) was used to obtain a superordinate factor representing emotional instability, consisting of high Neuroticism, moderate Openness, and low Extraversion, Agreeableness, and Conscientiousness. Persistent symptoms were measured using the head injury symptom checklist at six months post-injury; symptom severity (i.e., sum of all items) was used for further analyses. For patients, brain MRI was performed in the sub-acute phase (~1 month) post-injury. Following parcellation of rs-fMRI using independent component analysis, leading eigenvector dynamic analysis (LEiDA) was performed to compute dynamic phase-locking brain states. Main patterns of brain diffusion were computed using tract-based spatial statistics followed by PCA. No differences in phase-locking state measures were found between patients and HC. Regarding dMRI, a trend significant decrease in fractional anisotropy was found in patients relative to HC, particularly in the fornix, genu of the corpus callosum, anterior and posterior corona radiata. Visiting one specific phase-locking state was associated with lower symptom severity after mTBI. This state was characterized by two clearly delineated communities (each community consisting of areas with synchronized phases): one representing an

**Data Availability Statement:** Source data cannot be shared publicly because no formal consent was obtained to do so. Derivative data will be shared upon reasonable request to the corresponding

author. Data requests and other questions can be sent to the Medical Ethical Committee of the UMCG, Groningen, The Netherlands (metc@umcg.nl).

**Funding:** This research was supported by a Mandema stipend (reference number MA 18-02) from the University Medical Center Groningen to Harm Jan van der Horn. This funder had no role in study design, data collection and analysis, decision to publish, or preparation of the manuscript.

**Competing interests:** The authors have declared that no competing interests exist.

executive/saliency system, with a strong contribution of the insulae and basal ganglia; the other representing the canonical default mode network. In patients who scored high on emotional instability, this relationship was even more pronounced. Dynamic phase-locking states were not related to findings on dMRI. Altogether, our results provide preliminary evidence for the coupling between personality and dFNC in the development of long-term symptoms after mTBI.

## 1. Introduction

A mild traumatic brain injury (mTBI) can have a significant and long-lasting impact on a person's life, but the exact pathophysiological mechanism remains a conundrum for clinicians and scientists. Especially, untangling the influence of traumatic injury on the brain in the light of pre-existent factors, such as personality, emotion regulation and coping, has proven to be very challenging [1,2]. Advanced neuroimaging techniques, such as resting-state functional MRI (rs-fMRI), have improved our understanding of mTBI and its sequelae. The human mind is in constant motion, and so is the underlying functional architecture [3]. Considering this time-varying nature of the human brain, it is likely that dynamic functional connectivity (dFC) or its network analog dynamic functional network connectivity (dFNC) analysis of fMRI data is better suited for studying the complex pathophysiology of mTBI than its static counterpart [3,4]. There is a growing body of research on time resolved connectivity approaches to investigate mTBI, however, to our knowledge no studies have focused on dFC and personality [4–9].

Most measures of personality focus on the so-called Big Five personality traits, which are Neuroticism, Extraversion, Openness, Agreeableness, and Conscientiousness [10]. The strongest evidence for the role of personality in recovery after mTBI has been found for Neuroticism, which is positively correlated with persistent post-traumatic symptoms [11–13]. Neuroticism is the general tendency to focus on negative emotions and to worry, and therefore to experience higher stress levels. Other personality domains of the Big Five, namely Extraversion, Openness, Agreeableness, and Conscientiousness, have also been investigated, although the direction of the association with symptoms varies between studies [11–13]. Interestingly, studies in non-mTBI subjects have shown the existence of a hierarchical structure of the Big Five, which is the subdivision into superordinate personality traits based on a weighted combination of the Big Five [14]. For example, evidence has been found for a Two Trait structure (Big Two), consisting of one trait representing emotional (in)stability, and the other personal growth / positive emotionality [14,15]. As far as we know, this has not been examined in mTBI.

Various studies have examined brain activity and connectivity in relation to personality in healthy individuals [16–19]. Furthermore, dFNC has been found to be related to Neuroticism and Extraversion in patients with major depressive disorder [20]. Interestingly, cognitive networks that are related to Neuroticism, such as the default mode and salience network, have also been implicated in the pathophysiology of mild TBI [16,17,21,22]. To the best of our knowledge, the interaction between functional network connectivity and personality characteristics in mTBI has not been studied. It could be hypothesized that patients with adverse personality traits, for example characterized by high Neuroticism, are less able to recover from changes in functional connectivity resulting from the injury. It is not unlikely that perturbations in the neural circuitry that is involved in emotion regulation, which is closely related to

personality traits, may result in poor recovery [23,24]. This might also be related to micro-structural injury to underlying white matter tracts, although our previous work has not shown strong evidence for this [25]. Alternatively, pre-existent neural configurations of this circuitry might determine recovery after a mTBI. Altogether, it seems worthwhile to investigate dFNC in relation to personality in recovery after mTBI.

Over the last decade, the technical possibilities to measure dFNC have increased significantly. Studies using dFNC typically focus on changes in functional connectivity within reoccurring patterns of connectivity (i.e., states), changes in spending time in specific states, and/or changes in transition patterns between dynamic states in patients with mTBI relative to HC, as well as relationships with outcome [4,6,26]. The majority of work has focused on sliding window correlation (SWC) based approaches. This technique requires selecting a window size, an implicit filtering choice that optimizes the approach to a particular range of time-resolved changes. Initial studies use a single fixed window size, however more recent approaches, e.g., filterbanks, can cover an arbitrary range of frequencies [27,28]. Other approaches include those focused on computing connectivity patterns based on phase coherence, so called phase-locking states [29,30]. In this work we focus on a phase coherence approach called the leading eigenvector dynamics analysis (LEiDA) [29]. This technique captures recurrent patterns of phase locking between brain subsystems, by clustering the dominant eigenvectors of phase coherence obtained per time point. Here we apply this method to identify specific changes in dFNC in mTBI, which to our knowledge has not been done before.

The current exploratory study set out to investigate dFNC using the LEiDA method within the context of an independent component analysis (ICA) based parcellation. The following research questions were formulated: (1) Are phase-locking state measures different in patients with mTBI in the sub-acute phase (i.e., first weeks up to 3 months post-injury) relative to healthy controls (HC)?; (2) Is there an interaction between personality characteristics and phase-locking state measures explaining long-term recovery of patients with mTBI in terms of post-traumatic symptoms and functional outcome? Hereby we focus on personality features that have been associated with emotional instability [14]; (3) Are state measures related to microstructural changes as measured with diffusion magnetic resonance imaging (dMRI)? The latter question will provide insight in the association between dFNC and axonal injury (as reflected by abnormalities in diffusion metrics). We predicted that a personality profile associated with emotional instability would be related to poor long-term outcome, and that patients with a more emotionally unstable profile would spend more time in states that are associated with internal mentation (e.g., with a dominance of the default mode network) and would show fewer state transitions (i.e., more rigid/less flexible dFNC). We anticipated that this would be accompanied by little or no changes in dMRI measures, in accordance with our previous work [25].

## 2. Methods

### 2.1. Participants

In the period between January 17, 2020 and December 12, 2022, 27 patients with mTBI and 21 healthy controls were included in this neuroimaging study. This study was part of a larger prospective cohort study (AIM-TBI study, Dutch Trial Register/International Clinical Trials Registry Platform no. NL8484). Inclusion took place in the University Medical Center Groningen (UMCG), the Netherlands, which is a level 1 trauma center. Participants aged 18 years or older were included. Exclusion criteria were major neurologic or psychiatric comorbidity, admission for prior TBI (all severities included; this data was acquired during patient history at the Emergency Department and again via the two week questionnaire), drug or alcohol abuse, mental

disability, language barriers or illiteracy prohibiting understanding and completion of questionnaires, and contraindications for MRI. An additional exclusion criterion for HC was: any prior TBI. A mild TBI was defined by a Glasgow Coma Scale (GCS) score of 13–15, loss of consciousness ≤30 min and/or post-traumatic amnesia ≤24 hours, in accordance with the 1993 criteria of the American Congress of Rehabilitation Medicine [31]. All patients were seen by a Neurology resident during admission at the Emergency Department, and received head computed tomography (CT; based on the Dutch guideline for mild TBI [32]). The healthy control group was matched with the mTBI group based on age, sex and education.

The AIM-TBI study was approved by the by the Medical Ethical Committee of the UMCG (METc 2018/681), and all participants provided written informed consent. Study procedures were performed in accordance with the declaration of Helsinki.

## 2.2. Follow-up questionnaires

Longitudinal clinical data were acquired (while patients and HC were scanned only once). At two weeks post-injury, the NEO-Five Factor Inventory (FFI) was used to measure the Big Five personality traits [33]. This inventory consists of 60 items with each scored on a five point Likert scale ranging from 'strongly disagree' to 'strongly agree'. Total scores were computed for the following personality dimensions: Neuroticism, Extraversion, Openness to Experience, Agreeableness, and Conscientiousness. To obtain a main personality profile for every subject that is related to emotional instability [34], principal component analysis (pca; singular value decomposition) with varimax rotation was performed on the raw dimension scores (z-scored) using MATLAB's *pca* and *rotatefactors* functions. Subject scores on the first component (which corresponded with the superordinate trait *emotional instability;* explaining nearly half of the variance (47%)) were selected for further analyses.

At time of scanning, and at six months post-injury the head injury symptom checklist (HISC) was administered to patients; for HC the interval between first and second measurement was approximately six months [35]. This self-report questionnaire consists of items related to 21 frequently reported post-traumatic symptoms, which can be scored 0 (never), 1 (sometimes) or 2 (often) for both current as well as pre-injury situation. The severity of symptoms was calculated as the sum of all item scores. Only the current severity level was used, so that it could be compared with healthy controls. Twenty-five of all patients (89%) completed questionnaire at six months.

At six months post-injury, the Glasgow Outcome Scale Extended (GOS-E) questionnaire was also administered to patients with mTBI [36]. This eight-point scale measures outcome from (1) death to (8) full recovery. A dichotomy between complete recovery (score = 8) and incomplete recovery was used for statistical analyses (score < 8).

## 2.3. Imaging acquisition

MRI scans were made using a 3 Tesla Siemens MAGNETOM Prisma scanner (Siemens, Erlangen, Germany) equipped with a 64-channel SENSE head coil. Patients were scanned in the sub-acute phase post-injury (median of 30 days post-injury). A high-resolution transversal T1-MPRAGE volume was recorded for anatomical reference (repetition time (TR) 2300 ms; echo-time (TE) 2.98 ms; flip angle (FA) 9°; field of view (FOV) 240×256×176 mm; voxel size 1×1×1 mm). For resting-state fMRI, 400 volumes were recorded (with eyes closed) in descending order using a multi-band 4 sequence (TR 1600 ms; TE 34 ms; FA 70°; FOV 224×224×144 mm; voxel size 2×2×2 mm). For diffusion weighted imaging, single shot echo planar imaging was used in 64 diffusion directions (b-value = 1500 sec/mm$^2$, TR 5000 ms, TE 81 ms, FOV

220×220×132 mm, voxel size 2.2×2.2×2.2 mm). In addition, two volumes without diffusion weighting (b-value = 0 sec/mm$^2$) were recorded, one using anterior-posterior and one using posterior-anterior phase encoding direction. For 26 (93%) patients and all HC, also DWI data was recorded.

Lastly, a susceptibility weighted image (SWI; TR 39 ms; TE 31 ms; FA 15˚; FOV 227×250×141 mm; voxel size 0.8×0.8×0.8 mm), and T2-FLAIR image (TR 5000 ms; TE 386 ms; FOV 256×256×192 mm; voxel size 1×1×1 mm) were recorded. These images were assessed for traumatic lesions by an experienced radiologist (M.G.J.K.), who was blinded for group label and data regarding CT lesions.

## 2.4. fMRI analysis

In Supporting Information 1 a list of abbreviations and definitions regarding the neuroimaging analyses can be found. Raw imaging data were first converted to NIfTI using *dcm2niix* (v1.0.20210317) and then organized into BIDS format using custom built MATLAB software. Supporting Information 2 shows the fMRI processing pipeline. For preprocessing of the resting-state fMRI data, the *fMRIPrep* pipeline (v. 20.2.3) was used with slice timing correction, co-registration of the anatomical and functional images, "fieldmap-less" SyN-based susceptibility distortion correction, normalization to MNI space, and (non-aggressive) ICA-AROMA denoising (details of the processing steps can be found at https://fmriprep.org/en/latest/workflows.html) [37,38]. The outputs were inspected for errors. In addition, mean framewise displacement (FD) was computed for every subject, and compared between groups. There was no significant group difference (t = -0.73, P = 0.47).

Constrained spatial ICA was performed using the Group ICA of fMRI Toolbox (GIFT) implemented in MATLAB v. R2020a, with the *NeuroMark_fMRI_1.0* template (available at http://trendscenter.org/data and also within the GIFT toolbox at http://trendscenter.org/software/gift) as spatial prior, resulting in 53 components (further subdivided into subcortical (SC), auditory (AU), sensorimotor (SM), visual (VI), cognitive control (CC), default mode (DM), and cerebellar (CB) domains) for every subject [39–41]. Independent component (IC) time courses were further analyzed using the LEiDA method [29,42,43]. First, IC time courses were bandpass filtered using a 7$^{th}$ order Butterworth filter (0.04–0.07 Hz) using MATLAB *butter* and *filtfilt* function (which causes zero phase distortion). This bandwidth was chosen to reduce the influence of possible artifacts, and because the Hilbert transform works better (is better interpretable) on signals with a narrower frequency bandwidth [44]. The phase of IC time courses was estimated using MATLAB's *hilbert* function [45]. Seventeen volumes at the beginning and at the end were removed to account for edge effects that are inherent to filtering and Hilbert transformation (lowest frequency was 0.038 Hz, because of the slope of the filter, thus 1/0.038 = one cycle every 26.32 sec, thus 26.32/1.6 sec = 16.45 ≈ 17 volumes), leaving 361 volumes per subject. Subsequently, at each time point, phase coherence was calculated as the cosine of the difference in phases between two IC's, resulting in an IC × IC phase coherence matrix per time point. The first (leading) eigenvector ($V_1$) was computed per time point to extract the dominant connectivity pattern. A leading eigenvector at time point $t$ ($V_1(t)$) has elements that can be either positive or negative, representing the projection of the BOLD phase of the corresponding independent components onto $V_1(t)$. Based on positive and negative signs in $V_1(t)$, the brain is separated into two communities at time point $t$. The absolute values of the elements indicate how strongly components belong to a community. Since the sign of elements in $V_1(t)$ are arbitrary, unexpected sign flips can occur. To obtain consistency in eigenvector order and signs over time we used the *eigenshuffle.m* function created by John D'Errico (https://nl.mathworks.com/matlabcentral/fileexchange/22885-eigenshuffle). This function was

run on the (53 × 53 × 361) phase coherence tensor, resulting in a 53 × 361 matrix per subject containing all consistent first eigenvectors over time.

All of the subjects' time-varying leading eigenvector matrices were concatenated in the time direction, creating a 17,328 × 53 matrix. This matrix was analyzed using k-means clustering (using MATLAB's *k-means* function; with city block (also known as Manhattan) distance, 42 replicates, maximum iterations of 5000) to obtain phase-locking states. The optimal number of states was estimated using the Davies-Bouldin cluster validity index [46], which indicated an optimum of k = 5 (see Fig 1). The centroid leading eigenvectors ($V_c$) for every state contain elements (components) that have either positive or negative sign, corresponding with communities for a particular state. The phases of areas/components in the two communities within a specific dynamic phase-locking state are said to be 'locked' in space. The clustering algorithm assigns a state number (in our case ranging from 1 to 5) to each time point of a participant, meaning that the distance of the first eigenvector obtained from the phase-locking matrix at that time point to the specific state it has been assigned to is lower than the distance to the other states. This results in a vector of 361 state-assignment values for each subject. This vector was used to compute the following dynamic state measures: mean dwell time per state (i.e., mean time spent in a state before transitioning to another state), fraction of time spent per state (i.e., total number of time points spent per state divided by the total number of time points), number of state visits (i.e., how often a state is visited, a.k.a. visiting frequency), number of state transitions (i.e., how often a subject switches from one state to another), and state transition probabilities ($P\{X_{next}|X_{prev}\}$).

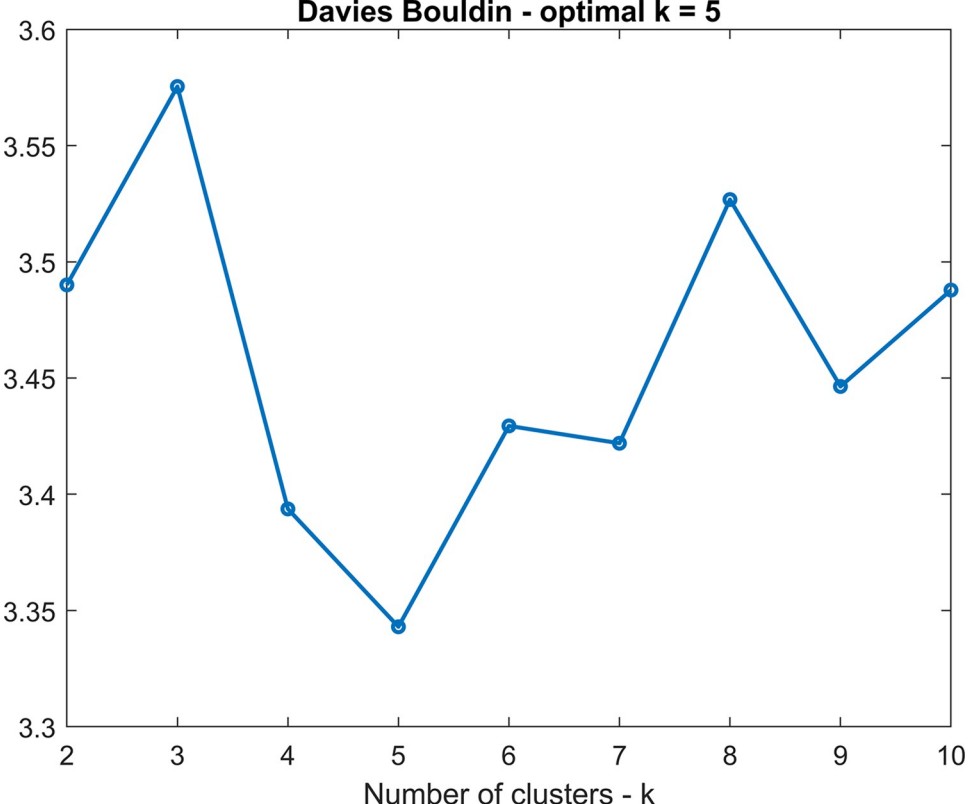

**Fig 1. Number of clusters determined using the Davies Bouldin index.** The optimal number of clusters is defined as the one that minimizes the ratio of the within cluster scatter to the between cluster separation. Plots was made using the scatter function implemented in Matlab.

## 2.5. DTI analysis

Supporting Information 3 shows the dMRI processing pipeline. Diffusion images were preprocessed using FMRIB Software Library (FSL; v. 6.0.5) [47]. Susceptibility distortions were corrected using the *topup* command on the B0 images with opposite phase-encoding direction. Subsequently, brain extraction was run on the distortion corrected B0 using the *bet* command, followed by correction of eddy currents and motion using the *eddy* command. Fractional anisotropy (FA), mean diffusivity (MD), and the three eigenvector and eigenvalue maps were generated using the *dtifit* command. The first eigenvector maps were inspected for correspondence with known anatomical tracts. Additional RD maps were computed using *fslmaths* (by taking the mean across the second and third eigenvalue maps).

For further analysis, FSL's tract-based spatial statistics (*TBSS*) was run using the ENIGMA pipeline (for protocol details we refer the reader to: https://enigma.ini.usc.edu/protocols/dti-protocols/ [48]. First, the individual subject's FA, MD, RD and AD maps were warped to the custom ENIGMA DTI template [49]. Subsequently, these maps were projected onto the ENIGMA DTI template skeleton. Individual skeletons were visually inspected for errors, and then parcellated into 46 areas using the JHU WM atlas. Average FA, MD, RD and AD for every region of interest (ROI) was computed. As mTBI is a heterogenous condition with respect to structural injury effects on the brain, pca (singular value decomposition) with varimax rotation was performed on z-scored ROI data using MATLAB's *pca* and *rotatefactors* functions, aimed at capturing the leading pattern of microstructural changes (an approach which also circumvents the need for a stricter correction for multiple comparisons, which might lead to type II errors in a condition with known heterogenic diffusion abnormalities). Thus, a Principal Component Analysis was performed on a $47 \times 46$ (subject $\times$ ROI) matrix for every diffusion metric (for FA, MD, RD and AD) separately. Subject scores on the first principal component were selected for further analyses (explaining 38% of the variance).

## 2.6. Statistical analyses

Statistical analyses of clinical data were performed using the Statistical Package for the Social Sciences (SPSS; IBM Corp. Released 2020. IBM SPSS Statistics for Windows, Version 27.0. Armonk, NY: IBM Corp). Group differences in continuous variables were analyzed using independent two sample t-tests; non-normally distributed data were tested using Mann-Whitney U Tests. Categorical data was analyzed using Chi-square tests. In case a questionnaire had multiple categories, results were considered significant at a Bonferroni corrected threshold (as for Big Five dimension subscores: $\alpha = 0.05/5 = 0.01$).

All further statistical analyses were performed using MATLAB's Statistics and Machine Learning Toolbox. Generalized linear models (MATLAB's *fitglm* function) were used to answer the main research questions.

For the assessment of group differences (mTBI vs. HC) in dynamic state measures (Y) the following model was used:

$$Y \sim 1 + Group + Age + Sex + Education$$

A similar model was used to assess group differences in DTI measures.

To examine the relationship between emotional instability and state measures, regarding long-term recovery (Y; i.e., symptoms or functional outcome) in the group of patients with mTBI, the following model formula was used:

$$Y \sim 1 + Emotional\ Instability*State\ Measure + Age + Sex + Education$$

The association between state measures (Y) and DTI measures in patients with mTBI was assessed using the following formula:

$$Y \sim 1 + TBSS\ PC1 + Age + Sex + Education$$

Here, TBSS PC1 indicates scores on the first principal component of all TBSS ROI data.

Age, sex and education were added as covariates in all analyses because of their relationship with functional networks, white matter integrity, and personality [50–53].

Histograms of dependent variables (Y) were visually inspected, and the appropriate distribution was selected accordingly (from Gaussian, poisson, gamma, and inverse gamma; binomial distribution was used for dichotomized GOS-E). In case a dependent variable did not sufficiently follow one of the known statistical distributions, the effect of a square root or log transformation was evaluated, and if satisfactory, applied to the data. The Akaike Information Criterion was used to compare and select (final) models. Model residuals were checked for normality, and for scedasticity using MATLAB's *plotResiduals* function and Shapiro-Wilk tests. In addition, variance inflation factors (VIF's) were computed for every independent variable in a model to measure the degree of collinearity (models with VIF's below 5 were accepted).

The overall alpha was set at 0.05. Bonferroni corrections were applied to the P-values of the overall regression models. To this end, we corrected for the number of states (k = 5; $\alpha = 0.05/5 = 0.01$) in case of state-specific measures (i.e., mean dwell time, fraction of time spent per state, number of state visits), and the total number of specific state transitions (k*k = 25; $\alpha = 0.05/25 = 0.002$) in case of state transition probabilities. Main and interaction effects of the variables of interest (i.e., all variables except for age, sex, and education) were checked for significance in case of a significant model.

## 3. Results

### 3.1. Demographics and participant characteristics

In Table 1 the general demographics and characteristics of both study groups are depicted. Patients showed a strong trend toward higher symptom severity at six months. Table 2 shows

**Table 1. General demographics and participant characteristics.**

| | mTBI (n = 27) | HC (n = 21) | Test statistic | P-value |
|---|---|---|---|---|
| Age, Mdn (range) | 36 (18–62) | 32 (19–70) | U = 282.5 | 0.98 |
| Sex, % female | 37 | 43 | $\chi^2 = 0.167$ | 0.68 |
| Education level, Mdn, range* | 6 (3–7) | 6 (5–7) | $\chi^2 = 1.773$ | 0.777 |
| Head Injury Symptom Checklist (severity) at time of scanning, Mdn (range) | 10 (1–33) | 5 (0–25) | U = 237.5 | 0.338 |
| Head Injury Symptom Checklist (severity) at 6 months, Mdn (range)† | 8 (0–36) | 6 (0–15) | U = 176.5 | 0.057 |
| Big Five subscores, Mdn (range) | | | | |
| Neuroticism | 26 (14–42) | 32 (17–42) | U = 552 | 0.023 |
| Extraversion | 43 (31–52) | 37 (29–49) | U = 744.5 | 0.086 |
| Openness | 36 (25–45) | 38 (24–45) | U = 612 | 0.307 |
| Agreeableness | 42.5 (37–50) | 47 (36–56) | U = 560.5 | 0.176 |
| Conscientiousness | 46 (33–53) | 46 (32–56) | U = 656.5 | 0.925 |

*Education level according to Verhage [54].

**Table 2. Clinical patient characteristics.**

| | |
|---|---|
| Glasgow Coma Scale score, Mdn, range | 14 (13–15) |
| Loss of consciousness, % | |
| *Yes* | 55.6 |
| *No* | 11.1 |
| *Possible* | 33.3 |
| Post-traumatic amnesia, % | |
| *Yes* | 92.6 |
| *No* | 7.4 |
| *Possible* | 0 |
| Injury Mechanism, % | |
| *Traffic—automobile* | 11 |
| *Traffic—scooter/moped* | 11 |
| *Traffic—bicycle* | 33 |
| *Fall* | 37 |
| *Other* | 8 |
| Time post-injury, days, Mdn (range) | 30 (17–48) |
| Traumatic lesions on CT, % yes | 22 |
| Traumatic lesions on MRI, % yes | |
| *Yes* | 22 |
| *No* | 63 |
| *Possible* | 15 |
| Hospitalized, % | 52 |
| Glasgow Outcome Scale Extended at 6 months, Mdn (range) * | 7 (4–8) |

*Abbreviations*: CT = computed tomography; MRI = Magnetic Resonance Imaging.

*Long-term outcome data was available for n = 25 patients.

the other clinical characteristics of the mTBI patient group. Of the 14 patients who were admitted to the inpatient ward, 11 (79%) were discharged the following day.

## 3.2. State measures in patients and healthy controls

Fig 2 shows the centroids for the five phase-locking states, depicted as the outer product matrices ($V_c^* V_c^T$), as well as spatial overlays based on a linear combination of maps of the independent components that have positive (red) or negative (blue) sign within the centroid $V_c$. Fig 3 shows an example of state transitions patterns for a patient in the mTBI group. Every subject (in both groups) visited each state at least once. Across all subjects and time points, 19.67% of time points was spent in state 1, 19.41% in state 2, 21.25% in state 3, 19.49% in state 4, and 20.18% in state 5.

There were no significant group differences in mean dwell time, fraction of time spent per state, number of state visits, number of transitions, nor state transition probabilities.

## 3.3. Interactions between state measures, personality, and outcome

The main principal component for personality that can be regarded as a measure of emotional instability was characterized by positive coefficients for Neuroticism (0.77) and Openness to experience (0.33), and negative coefficients for Extraversion (-0.45), Agreeableness (-0.13), and Conscientiousness (-0.28) (also see Supporting Information 4).

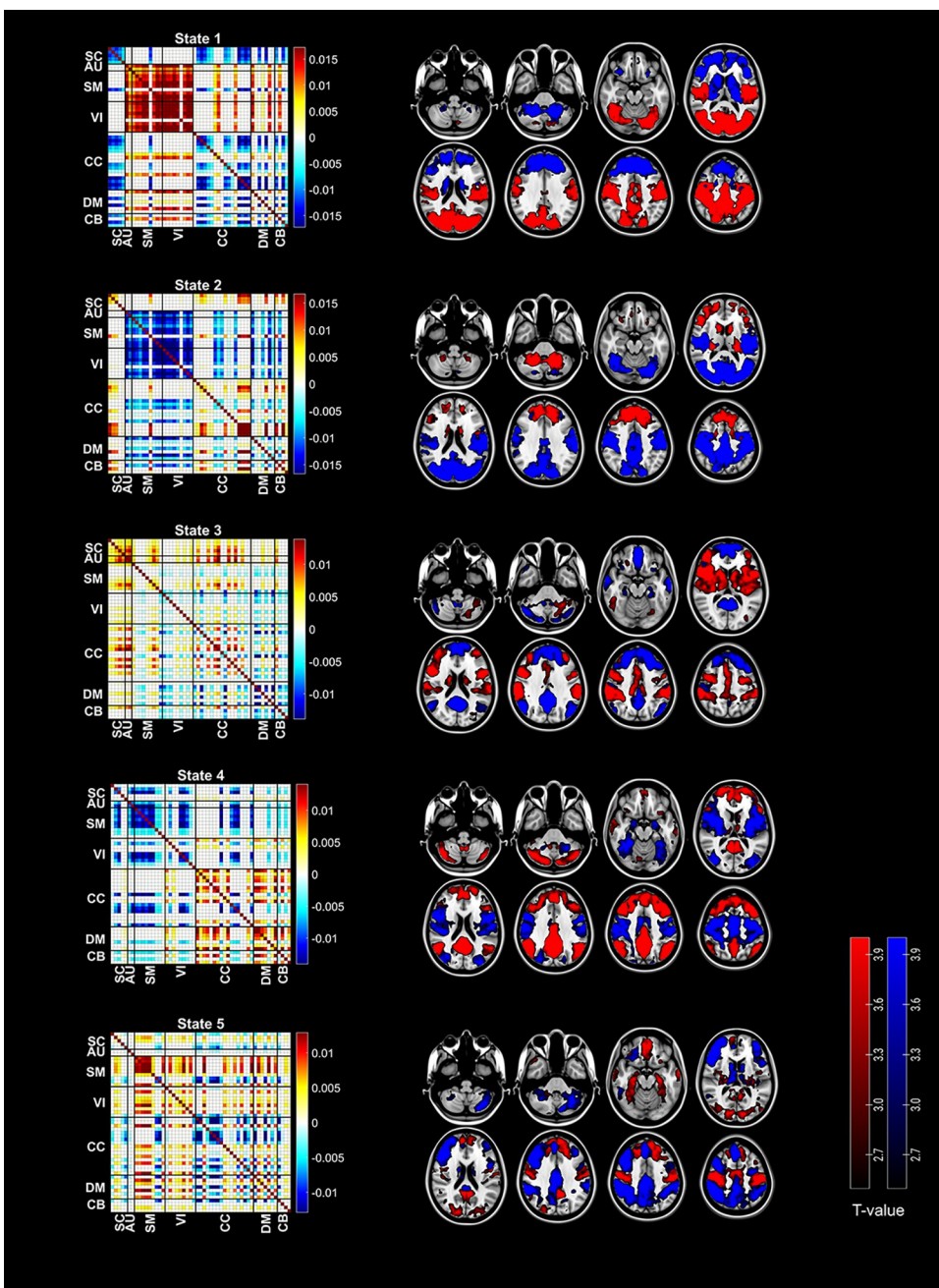

**Fig 2. Phase-locking states for the entire study sample.** On the left the $V_c * V_c^T$ (outer product) centroid matrices are shown with both communities color-coded: in red the connections between positive elements in $V_c$, and in blue the connections between negative elements (and zero for connections between elements of different communities). On the right side a weighted map is shown per state. This map consists of two layers: one in red containing the weighted positive (value in $V_c$ associated with each) independent components, and one in blue containing the independent components with negative value in $V_c$. The threshold was adjusted so that visualization was most optimal. Matrices were plotted using a custom Matlab script. Centroid images were created using a custom Matlab script, and were overlaid on 2D brain slices using Chris Rorden's MRIcroGL version 1.2.20210317.

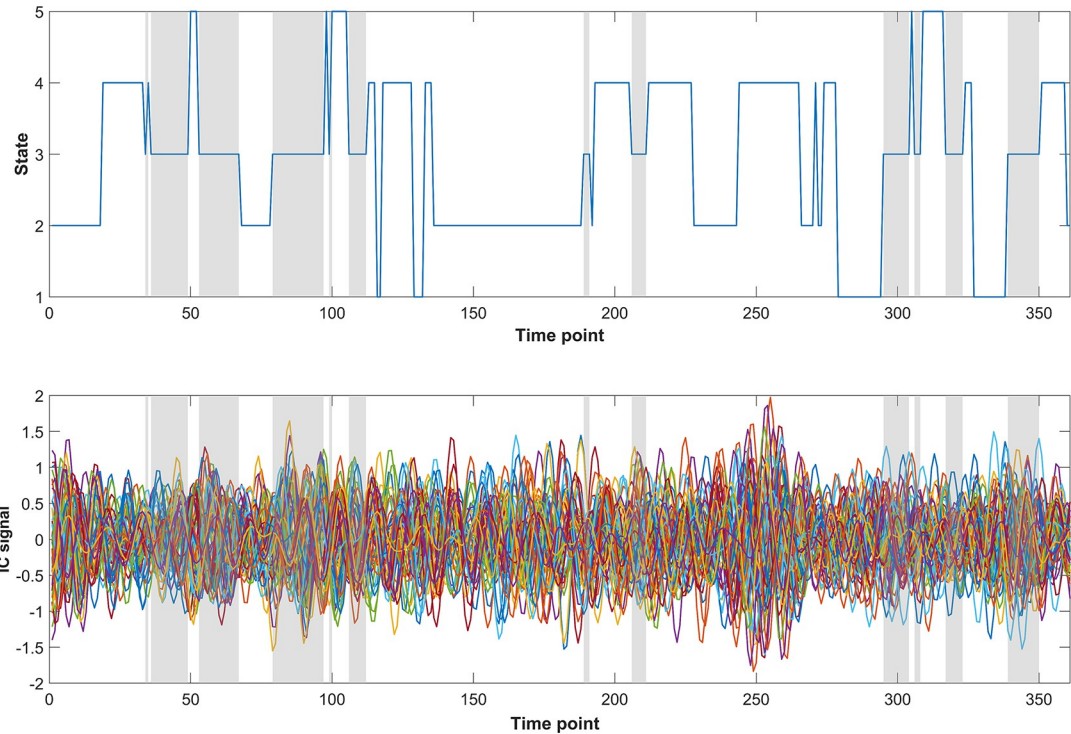

**Fig 3. The top figure illustrates state transitions over time for a patient in the mTBI group.** In the bottom figure the time courses of all 53 independent components are depicted. In shaded gray patches, visits to state 3 are shown. Plots were made using custom Matlab scripts.

Long-term symptom severity in the mild TBI group was significantly explained by the regression model including number of state 3 visits and emotional instability (F(8,16) = 7.26, $P_{corr}$ = 0.002). There were significant main effects for number of visits (t = -4.39, P = 0.0005) and emotional instability scores (t = 4.11, P = 0.0008) as well as a significant interaction between number of visits and emotional instability scores (t = -3.32, P = 0.004; Fig 4). Patients with higher emotional instability scores showed a strong negative relationship between number of state 3 visits and long-term symptom severity compared with patients that scored low.

In addition, the model including transition probability of state $2 \rightarrow 3$ and emotional instability significantly explained long-term symptom severity (F(8,16) = 8.73, $P_{corr}$ = 0.0005). Further exploration revealed main effects for state transition probability $2 \rightarrow 3$ (t = -5.45, P = 0.00005) and emotional instability scores (t = 5.54, P = 0.00004), accompanied by a significant interaction between transition probability of state $2 \rightarrow 3$ and emotional instability (t = -4.07, P = 0.0009).

Furthermore, the model including number of state transitions and emotional instability explained long-term symptom severity significantly (F(8,16) = 4.09, $P_{corr}$ = 0.008). The main effects for number of transitions (t = -2.19, P = 0.04) and emotional instability (t = 3.51, P = 0.003) were significant, as well as the interaction between these two factors (t = -2.8, P = 0.013).

No significant associations were found for long-term functional outcome (GOS-E = 8 vs GOS-E < 8).

Within the mTBI group, there was no significant relationship between the injury-to-scanning interval (in days) and state measures, when correcting for age, sex and education.

Within the HC group there were no significant results for number of visits to state 3, state transition probability $2 \rightarrow 3$, or number of transitions.

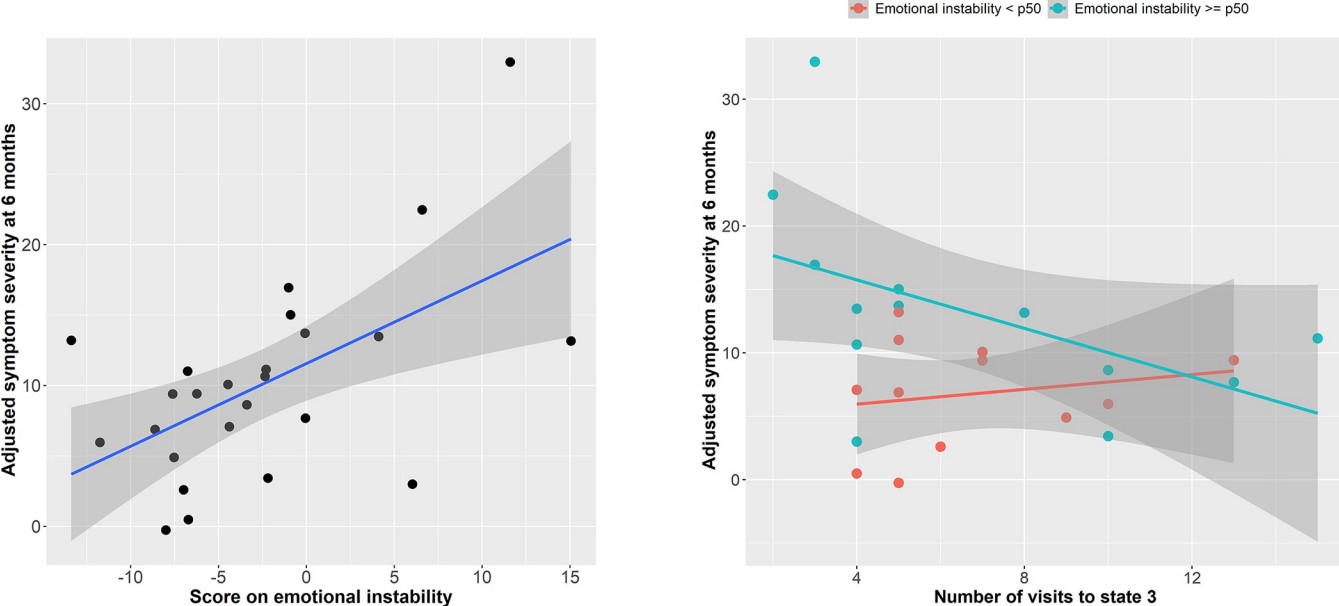

**Fig 4. Left figure shows the relationship between emotional instability scores and adjusted (fitted) long-term symptom severity in the mTBI group.** Right figure shows the interaction between emotional instability (low = blue datapoints and least squares line, high = red datapoints and line) and number of state 3 visits (x-axis) regarding adjusted long-term symptom severity (y-axis). Patients with higher emotional instability scores had a negative relationship between symptoms and number of visits compared to patients with low scores. Plots were made using ggplot2 (version 3.3.3) functions implemented in R Statistical Software (R Core Team, 2020). State 3 was characterized by one community consisting of insular, subcortical (basal ganglia), lateral prefrontal, supplementary motor, cingulate cortex, sensorimotor, and auditory areas, and another consisting mainly of areas that are associated with the canonical default mode network (with medial prefrontal, posterior cingulate, and precuneus as main areas) (see Fig 2). A description of the other states, and comparisons between states, can be found in Supporting information 5.

## 3.4. Relationship between state measures and DTI

Patients with mTBI scored lower on the first principal component for FA relative to healthy controls (F(6,40) = 3.93, P = 0.004; group: t = -1.82, P = 0.05). In Fig 5, it can be noted that the

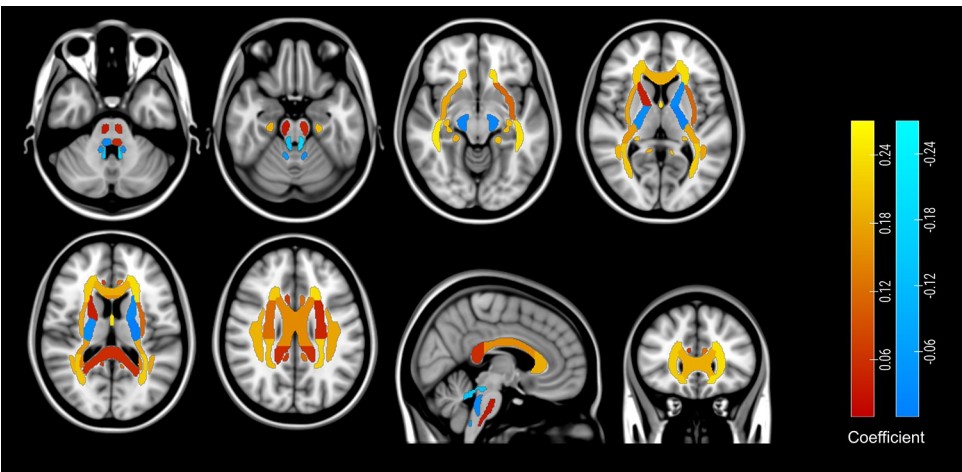

**Fig 5. Coefficients for the first principal component of TBSS FA ROI data.** Colors indicate the strength of coefficients; areas with brighter colors are more strongly associated with the first principal component. Images were created using a custom Matlab script, and were overlaid on 2D brain slices using Chris Rorden's MRIcroGL version 1.2.20210317.

highest principal component coefficients for FA were found for the fornix, genu of the corpus callosum, anterior and posterior corona radiata, and the lowest for the superior cerebellar peduncle. There were no significant models for MD, RD or AD.

No significant associations were found between FA first principal component scores and dynamic state measures.

## 4. Discussion

Sustaining a mild TBI can be a highly stressful event, both in the acute phase and at later time points post-injury. Therefore, someone's personality characteristics may determine the pattern of recovery to a great extent. In the current study we found preliminary evidence for an interaction between a personality trait representing emotional instability and dynamic state measures in the recovery process after mTBI. No differences in dynamic state measures were found between patients and healthy controls. We did find indications for diffusion differences in white matter between patients and healthy controls, but this was not related to dynamic state measures. Altogether, our results demonstrate that functional network dynamics related to personality might outweigh effects of the (physical) injury itself in the recovery of mTBI.

In the current study, no significant differences in phase-locking state dynamics were found between patients with mTBI and healthy controls. This finding is in accordance with our previous work on mTBI [4,22]. It is possible that connectivity changes related to the injury itself have already resolved at one month post-injury [55]. However, there is also work that suggests that changes in cerebral blood flow and functional connectivity can still be present, or become even more pronounced at one month post-injury compared to earlier time points [56,57]. Our dMRI analyses still showed some effects for fractional anisotropy, but not for the other diffusion metrics, which supports our fMRI findings.

A strong association was found between persistent posttraumatic symptoms and scores on a superordinate personality trait reflecting emotional instability, which was characterized by high Neuroticism and moderate Openness, as we hypothesized beforehand. Our results emphasize those of previous studies pointing to the role of Neuroticism in post-traumatic symptoms [11–13]. The emotional instability factor found in our study is compatible with the alpha-superordinate personality trait which was first reported by Digman et al., although we found a positive coefficient for Openness [15]. Other studies have reported a similar superordinate emotional stability factor, and in all studies this factor was characterized by high Neuroticism, but the degree to which the other dimensions contributed to this factor varied somewhat between studies [14,34]. Interestingly, a recent study has shown that a trait called Psychoticism from the Personality Disorder Inventory for the DSM-5, which is characterized by eccentricity and cognitive and perceptual dysregulation, was associated with chronic symptoms in patients with mild TBI [58]. This trait can be considered as the maladaptive end of the Openness dimension in the Big Five [59], which supports the moderate positive coefficient we found for Openness in the emotional instability trait.

Within the mTBI group we found a strong interaction between visits a particular pattern of brain-wide phase locking (state 3) and scores on the emotional instability component when it comes to long-term symptoms. For patients with higher emotional instability, a higher visiting frequency was particularly beneficial, meaning they could still recover by engaging in this state, which might have implications for developing future therapies. The results were not exactly as we predicted, because we expected that patients with more emotional instability, and more persistent symptoms, would spend more time in a state resembling the canonical default mode network. Instead, state 3 is characterized by two clearly delineated communities evolving in anti-phase with respect to each other. One represented an executive/saliency system, with a

strong contribution of the insulae and basal ganglia, but also of the lateral prefrontal cortex, the supplementary motor area, the anterior/mid cingulate cortex and the sensorimotor cortex. The other community represented the canonical default mode network. Previous studies on mTBI indicate that higher functional connectivity between the default mode and executive/salience networks is associated with more symptoms, possibly indicating a stronger need for inhibiting internally directed mental processes, leading to mental fatigue [21,60]. Additionally, in major depression disorder, higher connectivity between the executive and salience networks was found to be beneficial [61–63]. Therefore, the fact that in state 3 the executive and salience networks are in phase with each other while in anti-phase with the default mode network, could signify a more optimal network structure with respect to emotion regulation, which in turn could be associated with a more emotionally stable personality profile. A meta-analysis of task-based fMRI studies on emotion regulation has demonstrated the importance of many regions that were also found incorporated in state 3 in our study [64]. We found the insulae and basal ganglia to be strongly represented in the executive/saliency community. The insula and basal ganglia play an important role in the processing and regulation of emotions and associated behavior [65,66]. It could be hypothesized that a strong connection between the insula and basal ganglia is associated with more effective regulation of negative emotional stimuli (e.g., because of changes and difficulties experienced following the injury), which prevents rumination occurring in the presence of stronger connections between insula and default mode network.

As aforementioned, Neuroticism showed the highest coefficient within the emotional instability component. Neuroticism is associated with increased activation of (para)hippocampal areas, which are areas known to be associated with fear learning, which is the process of learning to predict whether stimuli are a threat [17]. The setpoint of this system might be higher in individuals with high Neuroticism, leading to a higher tendency to perceive (non-threatening) stimuli as threatening. The insula and anterior cingulate cortex (as part of the salience network) have also been found to be more active in high neurotic individuals during an avoidance vs. approach task, which further indicates a higher setpoint for threat perception and subsequent avoidance [19]. Furthermore, Neuroticism is associated with decreased activation of the basal ganglia (putamen/caudate) as well as the anterior and posterior cingulate cortex, which are areas associated with the adjustment of fear learning based on expectations [17]. This disbalance between learning and predicting can lead to uncertainty and stress, which can be alleviated by adequate cognitive control of emotions via engagement of lateral and medial prefrontal areas [17,64,67]. These areas are also involved in the executive/salience community of state 3. In neurotic individuals also a less efficient, and less modular intrinsic network organization has been found [16]. Altogether, these studies support the putative efficient network structure we observed for state 3. We also found an effect for number of state transitions, as we predicted, which we interpret as representing cognitive flexibility. The capacity to shift between brain states, and to arrive at states that are more efficient functional configurations (i.e., state 3) might result in better cognitive control of emotions, although we cannot prove causality here (the opposite might be true) [68]. Interestingly, within the HC group the results for personality, dynamic state measures and symptoms were null. A possible explanation may be that in non-stressful situations, there is less coupling between symptoms and personality. This theory is supported by previous literature by our group showing that the use of certain coping styles is not stable after mTBI [69].

Evidence for the relationship between personality and white matter structure is inconclusive [70,71]. In our study, we investigated whether there was an interaction between dMRI changes due to the injury and personality traits influencing outcome; we did not find evidence for such an effect. Altogether, it might be possible that dFNC configurations associated with

emotional instability are not induced by the injury, but rather reflect differences in pre-existent psychological vulnerability to stressful events.

This study has several strengths and limitations that require mentioning. We used multi-band fMRI which has a higher temporal and spatial resolution than non-multi-band sequences, and a higher statistical power due to a higher temporal degrees of freedom (and thus higher temporal signal-to-noise ratio). In addition, we used a validated preprocessing pipeline (fMRIPrep) [37], and included ICA-AROMA [38] to remove motion artifacts. Another strength is that we used multi-modal imaging techniques (fMRI and dMRI) to relate personality to both dynamic FNC as well as microstructural integrity. Furthermore, it could be argued that using ICA is a more biologically meaningful approach compared to using atlas-based parcellations, which is an additional strength of our study. The relatively small sample size (n = 27) is a limitation of our study. It has been shown that associations between psychological scores and fMRI data are difficult to reproduce in small samples, although there still is debate on this topic [72,73]. Furthermore, as far as we know, reproducibility of associations with dFNC has not been investigated. Therefore, our results need to be corroborated in larger N studies, preferably with a longitudinal design that allows for inferences about recovery regarding both clinical and imaging measures. Another limitation is the use of single-shell (one b-value) dMRI. Increasing evidence suggests that multi-compartmental models of diffusion are more sensitive to pathology of mTBI than traditional dMRI models [74]. Also, we used the GOS-E for measuring functional outcome. In patients with mTBI, GOS-E scores are heavily positively skewed. Therefore, we deemed it necessary to dichotomize this variable (into complete vs. incomplete recovery), and this may have resulted in decreased power and subsequent null findings. Lastly, our study population is representative of the civilian mild TBI population[75–77]. However, our sample is less comparable to populations at the milder end of the mTBI spectrum, such as patients with sports-related concussion [78].

To our knowledge, functional neuroimaging studies on personality and outcome after mTBI have not been published so far. Our study might be a starting point for further research, although larger samples are needed to corroborate our findings.

## Supporting information

**S1 File. List of abbreviations and definitions regarding fMRI and dMRI data analyses.**
(DOCX)

**S2 File. Resting-state fMRI pipeline.**
(DOCX)

**S3 File. Diffusion MRI pipeline.**
(DOCX)

**S4 File. First principal component from Big Five personality data.**
(DOCX)

**S5 File. Description of individual dynamic brain states.**
(DOCX)

## Acknowledgments

Thanks goes out to D.I. Larabi, B. Ćurčić-Blake, and J.R. Dalenberg for their valuable input during discussions regarding methodology, and A. Sibeijn-Kuiper, and J. Werdekker for their assistance during scanning.

## Author Contributions

**Conceptualization:** Harm J. van der Horn, Myrthe E. de Koning, Joukje van der Naalt.

**Data curation:** Harm J. van der Horn.

**Formal analysis:** Harm J. van der Horn, Marius G. J. Kok.

**Funding acquisition:** Harm J. van der Horn.

**Investigation:** Harm J. van der Horn, Myrthe E. de Koning.

**Methodology:** Harm J. van der Horn, Remco J. Renken, Joana Cabral.

**Project administration:** Harm J. van der Horn.

**Resources:** Harm J. van der Horn.

**Software:** Harm J. van der Horn.

**Supervision:** Harm J. van der Horn, Joukje van der Naalt.

**Validation:** Harm J. van der Horn.

**Visualization:** Harm J. van der Horn.

**Writing – original draft:** Harm J. van der Horn.

**Writing – review & editing:** Harm J. van der Horn, Myrthe E. de Koning, Koen Visser, Marius G. J. Kok, Jacoba M. Spikman, Myrthe E. Scheenen, Remco J. Renken, Vince D. Calhoun, Victor M. Vergara, Joana Cabral, Andrew R. Mayer, Joukje van der Naalt.

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
