## [Decision Letter · Decision Letter 0]

11 Sep 2023

PONE-D-23-25324Dynamic phase-locking states and personality in sub-acute mild traumatic brain injury: an exploratory studyPLOS ONE

Dear Dr. van der Horn,

Thank you for submitting your manuscript to PLOS ONE. After careful consideration, we feel that it has merit but does not fully meet PLOS ONE’s publication criteria as it currently stands. Therefore, we invite you to submit a revised version of the manuscript that addresses the points raised during the review process.

In particular, it would be nice if you could use more straightforward language to address the technical concerns as pointed out by the two reviewers.

We look forward to receiving your revised manuscript.

Kind regards,

Wang Zhan, Ph.D.

Academic Editor

PLOS ONE

Journal Requirements:

This research was supported by a Mandema stipend (reference number MA 18-02) from the University Medical Center Groningen to Harm Jan van der Horn. This funder had no role in study design, data collection and analysis, decision to publish, or preparation of the manuscript.

Reviewers' comments:

Reviewer's Responses to Questions

**Comments to the Author**

1. Is the manuscript technically sound, and do the data support the conclusions?

Reviewer #1: Yes

Reviewer #2: Yes

2. Has the statistical analysis been performed appropriately and rigorously? 

Reviewer #1: Yes

Reviewer #2: Yes

3. Have the authors made all data underlying the findings in their manuscript fully available?

Reviewer #1: No

Reviewer #2: No

4. Is the manuscript presented in an intelligible fashion and written in standard English?

Reviewer #1: Yes

Reviewer #2: Yes

5. Review Comments to the Author

Reviewer #1: The manuscript entitled "Dynamic phase-locking states and personality in sub-acute mild traumatic brain injury: an exploratory study" seems to be a compelling read that delves into the neural underpinnings of personality traits and their impact on the recovery from mild traumatic brain injury (mTBI).

The paper is well-structured and provides a comprehensive background on the existing literature, making it accessible even for those who may not be experts in the field. The methodology is robust, employing advanced neuroimaging techniques and statistical analyses to explore the complex relationship between personality traits and brain function in mTBI patients.

However, some of the technical jargon and statistical methods are a bit overwhelming. A more straightforward explanation of these methods could make the paper more accessible to a broader audience, who are interested in neuroscience but may not have an advanced understanding of the field.

The paper does a good job of acknowledging its limitations, including the relatively small sample size and the exploratory nature of the study. I appreciate the authors' caution in interpreting their findings, emphasising that these are preliminary results.

Overall, the paper is a fascinating exploration of how personality traits can influence the neural and symptomatic outcomes of mTBI. It opens the door for further research in this area, which could have significant implications for the treatment and management of mTBI.

Also, the following publication is closely related to mild traumatic brain injury (mTBI) and dynamic functional network connectivity, so it may fit in the point where the authors say 'Nevertheless, studies using time resolved connectivity

68 approaches to investigate mTBI are still relatively scarce' - Introduction. Paragraph 1 - because it slightly weak while knowing that there are studies for mTBI research using dFCN:

Aberrant Whole-Brain Transitions and Dynamics of Spontaneous Network Microstates in Mild Traumatic Brain Injury

- Authors: M. Antonakakis, S. Dimitriadis, M. Zervakis, A. Papanicolaou, G. Zouridakis

- Publication Date: January 15, 2020

- [DOI](https://dx.doi.org/10.3389/fncom.2019.00090)

Reviewer #2: This study is a captivating exploration of the influence of pre-existing personality traits, specifically neuroticism, on the recovery process following mild traumatic brain injury, employing dynamic functional connectivity and diffusion MRI techniques.

The manuscript is well written and organized, while the methods are generally well justified too. In particular, the finding of the correlation between phase-lock functional connectivity status and the persistence of post-traumatic brain injury symptoms in high-scored neuroticism mild TBI patients could be a meaningful contribution to this field.

In my assessment, this is a high quality study, which should come out for publication if the following relatively minor issues can be addressed.

General comments:

1. For the head injury symptoms in the healthy controls and mild TBI patients at scanning time, no significant difference is shown. I am a little curious if you have ever tried other symptom checklists, such as RIVERMEAD post-concussion symptom questionnaire. Could this explain, at least in part, the result of uncorrelated severity and diffusion metrics?

2. If possible, I would also like to see the correlation between phase-lock status and the head injury symptoms/ severity within healthy control group regarding to personality. Can this comparison be helpful to justify the “coupling between personality and dFNC in the recovery of mTBI”?

3. If not too much efforts required, I would be interested to see any static functional connectivity results and/or discussions, as a comparison to the present dynamic functional connectivity results.

Specific comments:

Abstract:

1. Line 55: the symptoms at 6-month post-injury only show the long-term severity and do not directly reflect the symptom improvement or recovery. Is it more appropriate to replace the phrase “recovery of TBI” with something like “long-term symptoms”?

Introduction:

2. Line 68: There are literature exist about dynamic functional connectivity application in traumatic brain injury, it would be helpful to include literature here.

Methods:

3. Line 129: This is a relatively small N of mTBI vs controls for such a dFNC analysis and post-injury symptoms correlation.

4. Line 291: The symptom severity at 6-month post-injury (not the symptom changes between scanning-time and 6-month post-injury) was selected for correlation analysis. Please justify your choice.

Results:

5. Line 334. Please give a description of each state just you did in line 377 for state 3.

6. Line 390. It is not clear about the long-term functional outcome.

7. Figure 4. High-resolution plots would be preferred.

6. PLOS authors have the option to publish the peer review history of their article (what does this mean?). If published, this will include your full peer review and any attached files.

Reviewer #1: No

Reviewer #2: No

---

## [Author Response · Author response to Decision Letter 0]

23 Oct 2023

Response to reviewers’ comments PONE-D-23-25324

We thank the reviewers for carefully examining our manuscript. Their comments have led to a substantial improvement of our work, for which we are grateful. Please find below a detailed response to every comment.

Reviewer #1: 

The manuscript entitled "Dynamic phase-locking states and personality in sub-acute mild traumatic brain injury: an exploratory study" seems to be a compelling read that delves into the neural underpinnings of personality traits and their impact on the recovery from mild traumatic brain injury (mTBI).

The paper is well-structured and provides a comprehensive background on the existing literature, making it accessible even for those who may not be experts in the field. The methodology is robust, employing advanced neuroimaging techniques and statistical analyses to explore the complex relationship between personality traits and brain function in mTBI patients.

Response: We thank the reviewer for the kinds words.

However, some of the technical jargon and statistical methods are a bit overwhelming. A more straightforward explanation of these methods could make the paper more accessible to a broader audience, who are interested in neuroscience but may not have an advanced understanding of the field.

Response: We acknowledge that there is a high density of technical jargon in the paper, that may be relatively difficult to parse for readers who do not have a specific background in these kind of methods. To facilitate the readability of our manuscript for a broader audience we decided to include a list of abbreviations and definitions regarding the imaging data analyses in the Supporting information. We hope that this list provides a guide for reading the manuscript. We referred to this list in the methods section.

The paper does a good job of acknowledging its limitations, including the relatively small sample size and the exploratory nature of the study. I appreciate the authors' caution in interpreting their findings, emphasising that these are preliminary results.

Overall, the paper is a fascinating exploration of how personality traits can influence the neural and symptomatic outcomes of mTBI. It opens the door for further research in this area, which could have significant implications for the treatment and management of mTBI.

Response: Thank you. We hope that the method presented in the current paper might be a novel starting point for further research on the consequences of mTBI.

Also, the following publication is closely related to mild traumatic brain injury (mTBI) and dynamic functional network connectivity, so it may fit in the point where the authors say 'Nevertheless, studies using time resolved connectivity

68 approaches to investigate mTBI are still relatively scarce' - Introduction. Paragraph 1 - because it slightly weak while knowing that there are studies for mTBI research using dFCN:

Aberrant Whole-Brain Transitions and Dynamics of Spontaneous Network Microstates in Mild Traumatic Brain Injury

- Authors: M. Antonakakis, S. Dimitriadis, M. Zervakis, A. Papanicolaou, G. Zouridakis

- Publication Date: January 15, 2020

- [DOI](https://dx.doi.org/10.3389/fncom.2019.00090)

Response: We agree, and thank the reviewer for this suggestion and included the paper, as well as others, as references to our introduction section. Furthermore, we rewrote this sentence in the introduction. 

Reviewer #2: 

This study is a captivating exploration of the influence of pre-existing personality traits, specifically neuroticism, on the recovery process following mild traumatic brain injury, employing dynamic functional connectivity and diffusion MRI techniques.

The manuscript is well written and organized, while the methods are generally well justified too. In particular, the finding of the correlation between phase-lock functional connectivity status and the persistence of post-traumatic brain injury symptoms in high-scored neuroticism mild TBI patients could be a meaningful contribution to this field.

In my assessment, this is a high quality study, which should come out for publication if the following relatively minor issues can be addressed.

Response: We thank the reviewer for the compliments. 

General comments:

1. For the head injury symptoms in the healthy controls and mild TBI patients at scanning time, no significant difference is shown. I am a little curious if you have ever tried other symptom checklists, such as RIVERMEAD post-concussion symptom questionnaire. Could this explain, at least in part, the result of uncorrelated severity and diffusion metrics?

Response: We agree with the reviewer that other questionnaires, such as the Rivermead, are of interest with respect to our findings. For the current study, however, we only acquired data using the HISC. We acknowledge the possibility that differences between items in questionnaires, but also other factors such as the way of scoring, may explain differences in correlations between imaging and clinical measures. We hope that our paper is a possible starting point for other researchers that use different questionnaires. 

2. If possible, I would also like to see the correlation between phase-lock status and the head injury symptoms/ severity within healthy control group regarding to personality. Can this comparison be helpful to justify the “coupling between personality and dFNC in the recovery of mTBI”?

Response: Based on the reviewer’s comment, we conducted the analyses for state 3, transitioning from state 2 to 3, and number of transitions in the HC group, and the results were null. We added this to our results section and included it in our discussion section. Our results may be explained by the fact that in or after stressful situations personality traits become more pronounced, as the individual is challenged to apply these traits. Although personality traits are considered stable, it has been shown by previous studies that this might not be the case for stressful situations, such as mTBI (1). In other words, in ‘stable’ situations there might be less coupling between symptoms and personality.

3. If not too much efforts required, I would be interested to see any static functional connectivity results and/or discussions, as a comparison to the present dynamic functional connectivity results.

Response: From a theoretical perspective it can be argued that personality is better described in terms of dynamic whole brain connectivity instead of static connectivity of single connections. This was the main incentive to focus on dynamic brain connectivity, and not on static functional connectivity.

We do agree that static functional connectivity may be of interest in mTBI. However, mTBI is a highly heterogeneous condition and results have been shown to vary significantly, especially in small N studies. The need for corrections for multiple testing add to the complexity of this matter. Future large N studies may focus on both dynamic functional connectivity and that of static connections (e.g., between prefrontal cortex and limbic regions) with regards to personality and outcome after mTBI.

Specific comments:

Abstract:

1. Line 55: the symptoms at 6-month post-injury only show the long-term severity and do not directly reflect the symptom improvement or recovery. Is it more appropriate to replace the phrase “recovery of TBI” with something like “long-term symptoms”?

Response: We agree and changed this in the abstract.

Introduction:

2. Line 68: There are literature exist about dynamic functional connectivity application in traumatic brain injury, it would be helpful to include literature here.

Response: Based on the comment of the reviewer, and that of the other reviewer, we updated our search and added some literature here, and rewrote this sentence in the introduction.

Methods:

3. Line 129: This is a relatively small N of mTBI vs controls for such a dFNC analysis and post-injury symptoms correlation.

Response: We agree that this is a small N study. Therefore we added the term ‘exploratory’ to our title. Based on the reviewer’s comment, we also added this term to the abstract.

4. Line 291: The symptom severity at 6-month post-injury (not the symptom changes between scanning-time and 6-month post-injury) was selected for correlation analysis. Please justify your choice.

Response: We chose for long term symptom severity because the imaging part of the study is cross sectional, and therefore we deemed that a prediction of delta symptoms was not fully justified. Also we aimed to limit the number of tests regarding relationships with outcome measures. We acknowledge the need for corroborating our results in longitudinal imaging studies to examine clinical recovery. We added a sentence on this topic to our limitations sections. 

Results:

5. Line 334. Please give a description of each state just you did in line 377 for state 3.

Response: A description for the other states and comparison between states was added to the Supporting Information. For the sake of readability we decided not to add this to the results section, but to refer to Supporting Information.

6. Line 390. It is not clear about the long-term functional outcome.

Response: Thank you for pointing this out. We clarified this sentence.

7. Figure 4. High-resolution plots would be preferred.

Response: High resolution plots were provided, but probably during the pdf conversion in the manuscript submission system low resolution images were created for review. We will make sure that the uploaded figures are in high resolution.

1. Scheenen ME, van der Horn HJ, de Koning ME, van der Naalt J, Spikman JM. Stability of coping and the role of self-efficacy in the first year following mild traumatic brain injury. Soc Sci Med [Internet]. 2017;1–7. Available from: http://linkinghub.elsevier.com/retrieve/pii/S0277953617301715

---

## [Decision Letter · Decision Letter 1]

4 Dec 2023

Dynamic phase-locking states and personality in sub-acute mild traumatic brain injury: an exploratory study

PONE-D-23-25324R1

Dear Dr. van der Horn,

We’re pleased to inform you that your manuscript has been judged scientifically suitable for publication and will be formally accepted for publication once it meets all outstanding technical requirements.

Kind regards,

Wang Zhan, Ph.D.

Academic Editor

PLOS ONE

Additional Editor Comments (optional):

Reviewers' comments:

Reviewer's Responses to Questions

**Comments to the Author**

1. If the authors have adequately addressed your comments raised in a previous round of review and you feel that this manuscript is now acceptable for publication, you may indicate that here to bypass the “Comments to the Author” section, enter your conflict of interest statement in the “Confidential to Editor” section, and submit your "Accept" recommendation.

Reviewer #1: All comments have been addressed

Reviewer #2: All comments have been addressed

2. Is the manuscript technically sound, and do the data support the conclusions?

Reviewer #1: Yes

Reviewer #2: Yes

3. Has the statistical analysis been performed appropriately and rigorously? 

Reviewer #1: Yes

Reviewer #2: Yes

4. Have the authors made all data underlying the findings in their manuscript fully available?

Reviewer #1: Yes

Reviewer #2: No

5. Is the manuscript presented in an intelligible fashion and written in standard English?

Reviewer #1: Yes

Reviewer #2: Yes

6. Review Comments to the Author

Reviewer #1: The authors have address all the comments. Before the final publication, the authors should proofread the final revised version to avoid any kind of typos etc.

Reviewer #2: (No Response)

7. PLOS authors have the option to publish the peer review history of their article (what does this mean?). If published, this will include your full peer review and any attached files.

Reviewer #1: **Yes: **Marios Antonakakis

Reviewer #2: No

---

## [Editor Report · Acceptance letter]

7 Dec 2023

PONE-D-23-25324R1 

Dynamic phase-locking states and personality in sub-acute mild traumatic brain injury: an exploratory study 

Dear Dr. van der Horn:

I'm pleased to inform you that your manuscript has been deemed suitable for publication in PLOS ONE. Congratulations! Your manuscript is now with our production department. 

Kind regards, 

on behalf of

Dr. Wang Zhan 

Academic Editor

PLOS ONE